# Percutaneous CT-Guided Bone Lesion Biopsy for Confirmation of Bone Metastases in Patients with Breast Cancer

**DOI:** 10.3390/diagnostics12092094

**Published:** 2022-08-29

**Authors:** Lucija Kovacevic, Mislav Cavka, Zlatko Marusic, Elvira Kresic, Andrija Stajduhar, Lora Grbanovic, Ivo Dumic-Cule, Maja Prutki

**Affiliations:** 1Clinical Department of Diagnostic and Interventional Radiology, University Hospital Center Zagreb School of Medicine, University of Zagreb, Kispaticeva 12, 10000 Zagreb, Croatia; 2Clinical Department of Pathology and Cytology, University Hospital Centre Zagreb, Kispaticeva 12, 10000 Zagreb, Croatia; 3Department for Medical Statistics, Epidemiology and Medical Informatics School of Medicine, University of Zagreb, Salata 12, 10000 Zagreb, Croatia; 4University North, 104 Brigade 3, 42000 Varazdin, Croatia

**Keywords:** breast cancer, bone metastasis, bone biopsy

## Abstract

We aimed to determine diagnostic accuracy of CT-guided bone lesion biopsy for the confirmation of bone metastases in patients with breast cancer and assessment of hormone receptor status in metastatic tissue. A total of 56 female patients with breast cancer that underwent CT-guided biopsy of suspected bone metastasis were enrolled in this retrospective study. Three different techniques were employed to obtain samples from various sites of skeleton. Collectively, 11 true negative and 3 false negative findings were revealed. The sensitivity of CT-guided biopsy for diagnosing bone metastases was 93.6%, specificity was 100% and accuracy was 94.8%. Discordance in progesterone receptor status and complete concordance in estrogen receptor status was observed. Based on our single-center experience, bone metastasis biopsy should be routinely performed in patients with breast cancer and suspicious bone lesions, due to the impact on further treatment.

## 1. Introduction

Breast cancer is the most frequently diagnosed malignancy among women worldwide [1]. Even though survival rates have increased in the recent years, there were still more than 130,000 deaths from breast cancer in Europe in 2018 [2]. Metastatic breast cancer was still the leading cause of death from all cancers in women. The majority of deaths from breast cancer are caused by metastases [3]. The most common site of breast cancer metastasis is bone, followed by lungs and liver. Bone metastases occur in approximately 70% of patients with advanced breast cancer. Aside from being the most prevalent site of breast cancer metastases, bone is also the initial site of breast cancer metastases in 26–50% of cases [1,4,5].

Although breast cancer metastases can already be present at the time of diagnosis, most often they develop after definitive treatment [3]. Moreover, metastatic breast cancer after therapy for early breast cancer tends to have a more aggressive tumor biology and subsequently a worse outcome compared with de novo metastatic breast cancer [6]. For example, the 5-year disease-specific survival of de novo metastatic breast cancer improved over time from 28% to 55%, whereas subsequent metastatic breast cancer decreased from 23% to 13%. Early detection of metastases is especially important to identify patients with localized metastatic disease, because for those patients, a more aggressive and multidisciplinary approach should be considered [7]. The monitoring of metastatic disease is crucial for evaluating response to treatment and preventing unnecessary toxicity from ineffective therapy. It includes periodic symptom assessments, physical examinations, laboratory tests, imaging methods and tumor markers. Recommended imaging methods for monitoring patients with metastatic breast cancer are computed tomography (CT), bone scintigraphy and positron emission tomography-computed tomography (PET/CT) [8,9].

The biopsy of suspicious lesions for metastatic disease, apart from confirming the presence of a metastasis, also has the advantage of allowing the assessment of hormone receptor status in metastatic tissue, which often differs from the main tumor, and may have a significant impact on the further course of treatment [10,11]. The biopsy is still not a common practice, although recent guidelines recommend biopsy at presentation and for the first recurrence of disease whenever possible [12]. The biopsy of bone lesions has the disadvantages of being technically demanding, requiring planning and having an unreliable outcome and, according to some guidelines, bone biopsy should be avoided [12,13,14].

The aims of this study were to determine the diagnostic accuracy of CT-guided bone lesion biopsy for the confirmation of bone metastases in patients with breast cancer, to identify variables which influence the quality of the histopathological specimen and to examine if estrogen receptor (ER), progesterone receptor (PR) and human epidermal growth factor receptor 2 (HER2) status of the metastasis differs from the immunohistochemical status of the primary tumor.

## 2. Materials and Methods

This retrospective study was granted approval by the institutional review board, which waived the requirement for informed consent. The electronic radiology information system was reviewed and 56 consecutive female patients (mean age 61.3 years; range 32–82 years) with pathologically proven breast cancer who underwent CT-guided biopsy of suspected bone metastasis between January 2018 and April 2021 were identified.

Percutaneous CT-guided biopsies of bone lesions were performed by an interventional radiologist according to the standard institutional procedure. Patients’ imaging data and laboratory tests were evaluated in advance to plan a safe and efficient biopsy. After the appropriate positioning of the patient on the CT gantry, a radiopaque marker was placed on the skin covering the expected biopsy location. A localized CT scan was then performed to confirm the selected site. The most appropriate CT slice was selected to determine the optimal puncture site and needle path and to measure lesion size and distance from the marker on the skin surface. Using the axial laser beam localizer built into the CT gantry, the puncture site was marked on the skin with an indelible pen. The radiopaque marker was removed and the skin was disinfected and draped. Local anesthesia using 1% lidocaine was applied. A small incision was made at the entry point and the coaxial needle was inserted along the predetermined path manually or using the battery-powered bone access system (Arrow^®^ OnControl^®^, Teleflex, Wayne, PA, USA). Repeat localized CT scans were performed to correct the needle direction and confirm the position of the needle inside the target lesion. Tissue specimens were obtained from each lesion using a 10 G biopsy needle (Tsunami Medical, Mirandola, Italy), 13 G needle (Teleflex, Wayne, PA, USA) or a semiautomatic 18 G needle (Bard Inc., Tempe, AZ, USA).

Two specimens of 20 mm maximum length for pathologic evaluation underwent fixation in 10% neutral buffered formalin. One-day decalcification was performed for all specimens with bone consistency using a mild-decalcifier solution (Osteosoft^®^, Sigma-Aldrich, Burlington, MA, USA). Core biopsy specimens from bone metastases that did not undergo decalcification were favored for immunohistochemistry to ensure that interpretation of ER, PR and HER2 was not compromised. 

Histologic evaluation was performed to confirm the metastatic disease and to verify adequate tumor cellularity for further immunohistochemical analysis. Adequate cellularity was defined as the presence of a few hundred tumor cells, even though ER, PR and HER2 testing are technically feasible on any number of demonstrable tumor cells. Immunostaining proteins were conducted on Ventana Ultraview Detection System with ER, PR and HER2 antibodies (Ventana, Tucson, AZ, USA, clones 6F11, 1E2 and 4B5, respectively), using positive and negative external controls. Results were dichotomized, with a positive result defined as 10% or more of tumor cell nuclei staining positively with any intensity.

Immunohistochemical analysis was performed to assess the status of ER, PR and human HER-2 in the metastasis and the results were compared to the primary tumor immunophenotype. The threshold for ER and PR positivity was 1%.

Tissue samples were classified into three categories according to their quality: optimal, suboptimal and low. Samples defined as suboptimal showed demonstrable tumor cells and the following features: low quantity (defined as cores < 5 mm in overall length), 5% or less tumor cellularity and necessity for detection of tumor cells by immunohistochemical analysis for cytokeratin. Samples insufficient for diagnosis or immunophenotype assessment were considered low quality.

Radiological and histopathological characteristics of suspicious bone lesions were analyzed. Histopathological characteristics of primary versus metastatic lesions were compared. Tissue specimen quality (optimal/suboptimal/low) depended on the method of biopsy (needle/semi-automatic needle/drill), site of biopsy, radiological characteristics (sclerotic/lytic/mixed lesions) and pathological characteristics (mostly bone/mostly soft tissue/mixed) of bone lesions was examined.

In case of negative histological findings, clinical follow-ups at six-monthly intervals, consisting of physical examination, laboratory tests and CT suspicious bone lesion, were required to differentiate between true and false negative findings.

Diagnostic accuracy for CT-guided biopsy of bone lesions for confirmation of metastases was calculated. Statistical analysis and plots were made by custom scripts written in Python 3.8 (Virginia, USA).

## 3. Results

A total of 58 lesions were biopsied in 56 patients. The following were the sites of biopsy in order of frequency: 31 (53.4%) iliac bone, 8 (13.8%) sternum, 6 (10.3%) vertebrae, 5 (8.6%) sacrum, 3 (5.2%) pubic bone, 3 (5.2%) rib, 2 (3.4%) femur. In total, 27 (46.6%) biopsies were performed using a drill, 25 (43.1%) with a needle and 6 (10.3%) using a semi-automatic needle. Histopathological analysis of biopsied bone lesions identified 44 bone metastases.

Using the CT-guided bone lesion biopsy as the bone metastases detection method, 11 true negative and 3 false negative findings were observed. No false positive findings were detected. Each of the false negative specimens was obtained using a drill. The sensitivity or number of true positives among all positive specimens of CT-guided biopsy for confirming bone metastases was 93.6%, and specificity, or the number of true negatives among all individuals without the metastases, was 100%. The overall accuracy of the procedure was 94.8%.

On CT examination, 14 (25.0%) patients presented with a solitary lesion and 42 (75.0%) patients had multiple lesions. Out of 58 lesions, 31 (53.4%) were osteolytic, 13 (22.4%) were sclerotic and 14 (24.1%) had combined features. Two out of three (66.7%) false negative findings presented as a lytic lesion, and one false negative finding appeared as a sclerotic lesion.

In six cases, the whole bone was involved. The remaining 52 lesions had an average diameter of 21.8 mm. The average diameter of false negative lesions was 26.0 mm.

Histopathological analysis revealed 14 negative and 44 positive findings. The quality and quantity of the sample depending on the method of obtaining tissue are presented in Figure 1. Representative sections of optimal and suboptimal samples were shown in Figure 2. All the false negative specimens were of low quality.

Analysis of the radiological characteristics of lesions revealed that lytic lesions exhibited the best quality, with 66.7% of optimal quality specimens. Sclerotic lesions had the lowest quality, with 28.6% of specimens being of low quality (Figure 3).

Samples collected from vertebrae had the highest proportion of optimal quality samples (83.3%). The site with the highest percentage of low-quality specimens was the pubic bone (33.3%). (Figure 4). All false negative samples were collected from different sites (iliac bone, pubic bone, sternum and thoracic vertebra).

There was not a statistically significant correlation between the sample quality and site of biopsy (chi2 = 4.010, *p* = 0.675), radiological characteristics (chi = 1.109, *p* = 0.574) or method of biopsy (chi2 = 0.284, *p* = 0.867). However, the pathological characteristics of the lesion were statistically significant (mostly bony/mostly soft tissue/mixed) (chi2 = 7.176, *p* = 0.028) (Table 1).

According to the prior assessment of the primary tumor, 47 samples were ER positive, 4 were ER negative and 7 samples were ineligible for analysis. Immunohistochemical evaluation of suspicious bone lesions revealed 40 ER positive and 2 negative findings. A total of 16 samples were unsuitable for analysis. All 38 paired primary tumor metastasis samples were concordant for ER status. In terms of PR status, 38 primary tumors were positive, 13 were negative and 7 could not be assessed. In total, 23 bone samples were PR positive, 17 were negative and 18 were inadequate for analysis. Of 36 paired samples, 24 (66.7%) were concordant and 12 (33.3%) were discordant regarding PR status. Moreover, 4 primary tumors were Her-2 positive, 48 were negative and 6 samples could not be assessed. One bone lesion was Her-2 positive, 40 were negative and 17 were unsuitable for analysis. Discordance for Her-2 status was noted in 1 (2.6%) case out of 38 paired samples (Figure 5).

No biopsy-related complications were reported.

## 4. Discussion

Breast cancer metastasis could possibly occur after a longer period, but most recurrences occurred in the first 5 years after diagnosis. Therefore, the long-term survivors have greater chance to be ‘cured’ [15]. Metastatic breast cancer is so far considered as an incurable disease, but survival improvements have been reported with appropriate therapeutic strategies [16]. In general, tumors characterized as HR positive are considered as less aggressive phenotype with higher overall survival rate compared to HR negative tumors. Triple-negative breast cancer-lacking hormone receptors and HER2 receptor had higher recurrence rate and subsequent poor outcome [17]. Presence of ER receptors enabled a reachable therapeutic target for anti-estrogen drugs, improving survival among these patients [18]. However, ER-positive tumors showed the capability to develop hormone resistance and progression to metastases, even with anti-estrogen therapy. Additionally, ER and PR status of metastatic tissue was revealed to be discordant with receptors of primary tumor [19,20,21].

Patients with a newly diagnosed or recurrent metastatic breast cancer should have a biopsy, if technically feasible, to confirm histology and to re-assess ER, PR and HER2 status [13]. In the previous studies, the discordance rates in ER, PR and HER2 status were found to be 16, 40 and 10%, respectively, and the biopsy results altered management in 15.9% of patients [22,23]. In our study, there was a discordance in PR status, while there was no discordance in ER status. HER2 status showed discordance in only one of the analyzed samples. Tumor heterogeneity is a known characteristic of breast cancer and could be an important signal for change in receptor status. Initial biopsy of a primary tumor could be performed on areas with the expression of a particular receptor, while other areas will, in the future, become the true source of tumor cells that grow and give rise to the metastasis [24]. Alternatively, the change in tumor receptor status could be caused by genetic changes either due to genomic instability of the tumor cells or because of primary tumor treatment, as suggested by Lindström and collaborators [25].

According to European Society for Medical Oncology (ESMO) guidelines, biopsies of bone metastases should be avoided, whenever possible, due to the limited possibilities of biomarker detection in decalcified tissue [13]. However, a recent study has suggested that decalcification by EDTA minimally affects receptor expression results, therefore enabling reliable assessment of receptor status in samples gained by bone biopsies [26].

Bone metastasis biopsy was shown to be important in the reassessment of biological features, thus having an impact on the choice of the further treatment [27]. Despite extensively described evidence and current recommendations, the acquisition of samples from metastatic tissue is still not a routine practice. Consequently, therapeutic decisions for metastatic breast cancer are based on the features of the primary tumor. Invasiveness of the procedure and the unreliable outcome of the biopsy, especially when performed at complex visceral sites, are among the main reasons for the avoidance of routine biopsies. Improvements in interventional radiology techniques enabled safe access to majority of metastatic sites [28]. The available literature has suggested that the morbidity and complication rate associated with bone biopsy are minimal when performed in experienced centers. In the trial designed by Amir and collaborators, most participants (89%) recommended metastatic biopsy to other patients [23].

In general, CT-guided biopsy of bone lesions in cancer patients allows for a final diagnosis in 94% of cases, a specimen longer than 1 cm may lead to a significant result in terms of adequacy and sensitivity and negative biopsies with positive positron emission tomography or magnetic resonance imaging and a specimen shorter than 1 cm should be repeated to avoid a false negative result [Monfardini]. We reported a similar percentage of false negative findings. Among 58 lesions analyzed in our study, 3 were shown to be false negative. Two out of three were lytic lesions and one was sclerotic, all of them collected from different sites. The samples collected from vertebrae had the highest proportion of optimal quality samples, probably due to the high percentage of trabecular bone which allows better sampling. In contrast, the site with the highest percentage of low-quality samples was pubic bone, which might be explained by the bone here being stiffer. All three false negative samples were collected using a drill. Based on our study, results obtained by needle and semi-automatic needle performed biopsy did not have false negative cases. Further studies and larger cohorts are needed for confident conclusions about optimal biopsy technique. Techniques described in the study performed by Monfardini and collaborators support the usage of needles and avoidance of drills [29]. Roberts et al. aimed to detect the most appropriate needle for bone biopsy by evaluating eight different needles on an animal lumbar spine model with maximum specimen length limited to 1 cm [30]. Parameters for quality assessment were degree of fragmentation, trabecular distortion and crush artifact. Needles showed great variability in the performance. However, the Trap-lock 11G needle system was considered the best from the perspective of specimen quality. In our study, we decided to utilize two different sizes, depending on technique: 10G needle and 18G semi-automatic needle. One study focused on samples longer than 1 cm revealed significant correlation between the length of the specimen and both inadequate and false negative results [29]. According to them, the longer the specimen is, the lower the probability for false negative or inadequate results. The sample is considered adequate when the receptor assessment is feasible and reliable. The overall rate of complications was very low in this study (3.5%), which was similar to our study (no reported complications) and is a direct consequence of enhanced biopsy techniques.

The difficulties of sampling tissue from bone are so far well described. Biopsies of bone metastases and bone marrow rarely yield enough tissue for robust molecular biology studies using clinical samples. The proposed solution is to collect larger biopsy specimens or use improved RNA extraction techniques [31]. In our study, we successfully utilized different techniques for sampling. We, therefore, suggest that both needles (10G and semi-automatic 18G) can be employed, depending on the biopsy site and radiologist’s experience.

Collectively, CT-guided bone biopsy in breast cancer patients with suspicious metastasis is safe and should be considered as an important diagnostic tool, due to significant implications on further treatment planning. Based on our single-center experience, the sensitivity of CT-guided biopsy for detecting bone metastases was 93.6%, specificity was 100% and accuracy was 94.8%. Marked discordance in progesterone receptors and complete concordance in estrogen receptor was shown. Cases of negative biopsy result and suspicious PET and/or MR findings should be additionally assessed and considered for re-sampling. The main limitations of our study were its retrospective nature and limited number of patients. Therefore, further study should include more patients and have a prospective design.

## Figures and Tables

**Figure 1 diagnostics-12-02094-f001:**
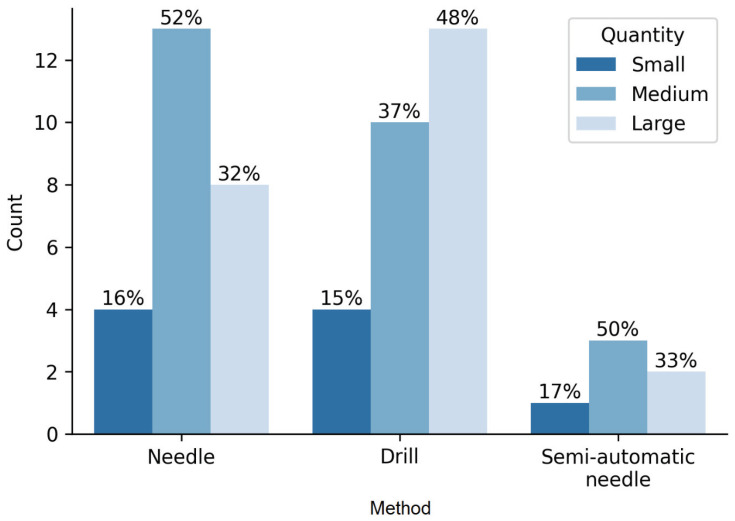
Characteristics of tissue samples and methods of collection.

**Figure 2 diagnostics-12-02094-f002:**
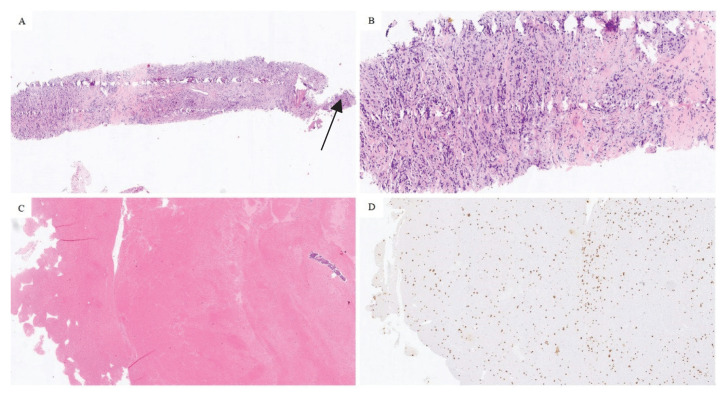
Optimal tumor quantity in a bone biopsy specimen, hematoxylin and eosin staining, magnification 20× (**A**). Note small osseous fragment (arrow). Higher-power view of tumor cells, same section, magnification 100× (**B**). Suboptimal specimen quality, most of the material is in the form of a clot with scattered tumor cells, hematoxylin and eosin staining, magnification 20× (**C**). Tumor cells were apparent only with cytokeratin AE1/AE3 staining, magnification 20× (**D**).

**Figure 3 diagnostics-12-02094-f003:**
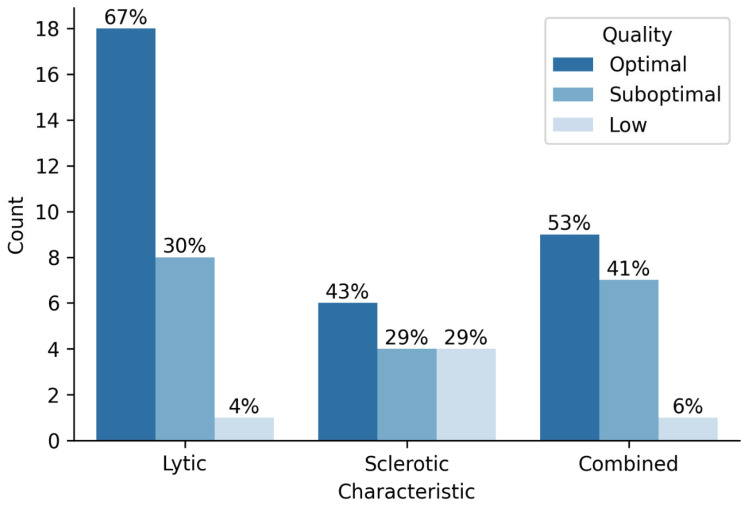
Sample quality and radiological characteristics of lesions.

**Figure 4 diagnostics-12-02094-f004:**
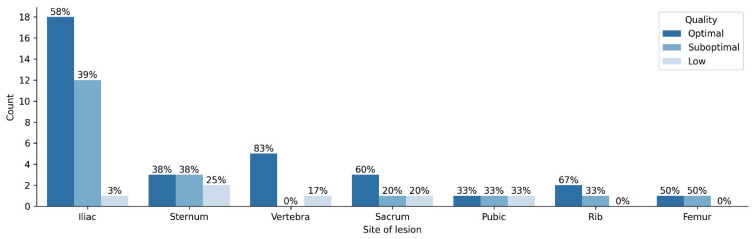
Sample quality and site of lesion.

**Figure 5 diagnostics-12-02094-f005:**
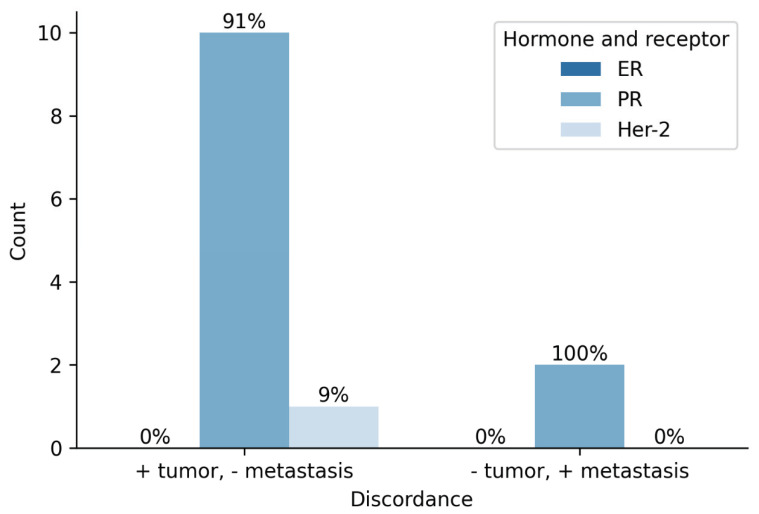
Status of hormone and Her2 receptors in primary tumor and metastasis.

**Table 1 diagnostics-12-02094-t001:** Sample quality and pathological characteristics of the lesion.

	Mostly Bone	Mostly Soft Tissue	Mixed
Optimal	9	14	11
Suboptimal	11	12	1

## Data Availability

Data used for analysis is contained within the article.

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
