# Peer review of "Percutaneous CT-Guided Bone Lesion Biopsy for Confirmation of Bone Metastases in Patients with Breast Cancer"

_diagnostics, 2022, doi:10.3390/diagnostics12092094_

Round 1

Reviewer 1 Report

This manuscript reported the results of CT-guided bone lesion biopsy for detecting breast cancer-associated bone metastasis from 58 lesions of 56 patients. The data support the usefulness of CT-guided bone lesion biopsy, which has been well recognized in this diagnostic area. Beyond the supportive information, not much new information is presented and novelty is low.

Major

·         Materials and methods: since this focuses on the performance of CT-guided bone lesion biopsy, it is recommended to expand the description of materials and methods and include a standard procedure as well as any deviations from the standard procedure.

·         Sensitivity, specificity, and accuracy: these terms should be explained. since the sample number is 58 (2 digits), the three-digit values should be converted to two-digit values.

·         Comparison to reported studies using CT-guided bone lesion biopsy: Since the procedure is widely employed, the authors may present the comparison of their results with the published studies.

·          Other methods: It is recommended to evaluate the performance of the CT-guided procedure in comparison with other methods of detecting bone metastasis.

Minor

·         Plots rather than tables: It is recommendable to use plots/charts to present the results rather than showing the numbers in tables.

Author Response

We thank reviewer for the constructive criticism which has significantly improved the quality of our revised manuscript.

1) Detailed description of CT-guided bone lesion biopsy was added to materials & methods section of the manuscript (lines 76 – 94).

2) Explanation of terms sensitivity, specificity and accuracy were incorporated in the revised manuscript. Changes are marked in lines 139 - 145.

3) According to European Society for Medical Oncology (ESMO) guidelines the acquisition of sample from metastatic tissue is still not a routine practice. Therefore, our aim was, collectively with other published studies (references 28-30), suggest that CT-guided bone biopsy in breast cancer patients with suspicious metastasis is safe and should be considered to become a part of protocol, due to significant implications on further treatment plan.

4) Comparison of the CT-guided procedure with other methods of detecting bone metastasis is an excellent point. Bone metastases can be detected by utilizing different imaging modalities.

Although metastases can be detected on plain films, most often in the patients presenting with symptomatic bone pain or pathological fractures, plain films are not suitable as a screening test for detecting metastases due to the low sensitivity (44-50%) for metastasis detection. (Hamaoka T, Madewell JE, Podoloff DA, Hortobagyi GN, Ueno NT: Bone imaging in metastatic breast cancer. J Clin Oncol 2004; 22: 2942–53)

Bone scintigraphy (sensitivity 86%, specificity 81%), CT (sensitivity 73%, specificity 95%), PET-CT (sensitivity 90%, specificity 97%) and MRI (sensitivity 91%, specificity 95%) are more appropriate techniques for detecting bone metastases. [Yang HL, Liu T, Wang XM, Xu Y, Deng SM: Diagnosis of bone metastases: a meta-analysis comparing (1)(8) FDG PET, CT, MRI and bone scintigraphy. Eur Radiol 2011; 21: 2604–17]

However, newly detected bone lesion may not be related to the known primary tumor and biopsy of suspected metastases enables accurate diagnosis of metastatic disease. Considering that the biopsy is used for the confirmation, and not for detection of metastatic lesions (in radiologic terms), the title of the paper was modified accordingly.

5) As you recommended, we included plots instead of tables, so now we have Figure 1, 3, 4, 5 instead of Table 1, 2, 3, 5.

We hope our manuscript will be suitable for publishing in your distinguished Journal and of interest to the broad audience.

Reviewer 2 Report

Dear Authors,

Your paper is very interesting but a bit confusing. Since you aimed to retrospectively analyze the diagnostic performance of CT-guided bone biopsy in metastatic breast cancer, I think that this aspect should be better explained. Indeed, you showed high specificity and accuracy of CT-guided biopsy in detection of breast cancer metastatic lesions but there is no description of how you gained these results. In particular, we don’t know anything else that the presence of bone lesions as for the characteristics of the enrolled patients. It could be useful to add some images and at least a briefly description of your findings, if it is possible.

Furthermore, you focused on the evaluation of hormonal status of the metastatic lesions, stating that a possible discordance between primary tumor and metastatic lesion could influence the therapy. Since it is widely recognized that patients with metastatic lesions after therapy for early breast cancer have a worse outcome compared with a de novo metastatic disease, I think that to make conclusions more strong, you should better elaborate on this aspect, showing for example, how many patients belonged to the two different category and if, according to your findings, the patients received a different treatment or not.

An extensive revision of the English language is necessary (e.g. lines 267-268, please, correct verb tenses).

Lines 27-36 are redundant and need to be better organized to avoid repetitions.

Please, correct references in according to journal layout.

Kind regards

Author Response

We thank reviewer for the constructive criticism which has raised important questions and significantly improved the quality of our revised manuscript.

1) Using the CT-guided bone lesion biopsy as bone metastases detection method, 11 true negative and three false negative findings were observed. No false positive findings were detected. Each of the false negative specimens was obtained using a drill. The sensitivity or number of true positives among all positive specimens of CT-guided biopsy for detecting bone metastases was 93.6% and specificity, or the number of true negatives among all individuals without the metastases was 100%. The overall accuracy of the procedure was 94.8%

2) Regarding the question about presence of bone lesions among our patients, we included new figure in manuscript, called Figure 2, which comprised figures of optimal (20x and 100x magnification) and suboptimal (20x magnification) tumor quantity in a bone biopsy specimen, stained by hematoxylin and eosin, and cytokeratin AE1/AE3 staining for tumor detection

3) In our study we did not concentrate on the stage of the disease. Discordance in progesterone receptor status between primary tumor and bone metastasis was observed in 9 out of 30 (30%) patients who previously underwent therapy and in 3 out of 6 (50%) patients with initially metastatic disease. We did not find these results to be significant given the small number of patients in the second group. A single case of discordance regarding HER-2 status was observed in the patient who previously underwent therapy.

4) According to your suggestions regarding the English language, verb tenses were corrected and English spell check applied (marked as track changes).

We hope our manuscript will be suitable for publishing in your distinguished Journal and of interest to the broad audience.

Round 2

Reviewer 1 Report

The authors responded sufficiently to the comments to the previous version. Thank you.